# Aspirin Resistance in Vascular Disease: A Review Highlighting the Critical Need for Improved Point-of-Care Testing and Personalized Therapy

**DOI:** 10.3390/ijms231911317

**Published:** 2022-09-26

**Authors:** Hamzah Khan, Omar Kanny, Muzammil H. Syed, Mohammad Qadura

**Affiliations:** 1Division of Vascular Surgery, St. Michael’s Hospital, Toronto, ON M5B 1W8, Canada; 2Keenan Research Centre for Biomedical Science, Li Ka Shing Knowledge Institute of St. Michael’s Hospital, Toronto, ON M5B 1T8, Canada; 3Department of Surgery, University of Toronto, Toronto, ON M5T 1P5, Canada

**Keywords:** aspirin, resistance, platelet, aggregation, point-of-care, antiplatelet, vascular

## Abstract

Aspirin resistance describes a phenomenon where patients receiving aspirin therapy do not respond favorably to treatment, and is categorized by continued incidence of adverse cardiovascular events and/or the lack of reduced platelet reactivity. Studies demonstrate that one in four patients with vascular disease are resistant to aspirin therapy, placing them at an almost four-fold increased risk of major adverse limb and adverse cardiovascular events. Despite the increased cardiovascular risk incurred by aspirin resistant patients, strategies to diagnose or overcome this resistance are yet to be clinically validated and integrated. Currently, five unique laboratory assays have shown promise for aspirin resistance testing: Light transmission aggregometry, Platelet Function Analyzer-100, Thromboelastography, Verify Now, and Platelet Works. Newer antiplatelet therapies such as Plavix and Ticagrelor have been tested as an alternative to overcome aspirin resistance (used both in combination with aspirin and alone) but have not proven to be superior to aspirin alone. A recent breakthrough discovery has demonstrated that rivaroxaban, an anticoagulant which functions by inhibiting active Factor X when taken in combination with aspirin, improves outcomes in patients with vascular disease. Current studies are determining how this new regime may benefit those who are considered aspirin resistant.

## 1. Introduction

Atherosclerotic disease is the leading cause of death worldwide [1,2]. Atherosclerotic plaque, often found in the intimal layer of arteries, can significantly reduce blood flow and nutrient/oxygen delivery to peripheral muscles and organs [3]. In severe cases, atherosclerotic plaques can rupture and cause thrombus formation, which can travel downstream and completely block arteries, leading to end-organ damage and cell/organ death [4].

Depending on the severity and anatomical location of the atherosclerotic plaque, patients can develop different progressive disorders (Figure 1). Peripheral artery disease (PAD) occurs when plaque buildup is found in the intima of peripheral arteries, usually within the lower extremities, and is associated with pain during ambulation, known as “intermittent claudication” [5]. PAD is associated with a significant increase in the risk of myocardial infarctions (MI), cerebral vascular accidents (CVA), limb amputation and cardiovascular related death [6]. Carotid artery stenosis (CAS) is another common manifestation of atherosclerotic disease. CAS is specifically defined as the buildup of plaque within either the internal, or external carotid artery, and can lead to transient ischemic attacks, or CVA, where there is a lack of blood flow to the brain [7].

Due to increased flow rates, shear stress, turbulent blood flow, and endothelial injury that occurs at the location of atherosclerotic plaque, an environment for inappropriate platelet activation is created [8,9,10,11]. Several studies report that patients with PAD and CAS have significantly higher platelet activity when compared to healthy controls [12,13,14,15,16,17,18,19,20]. CAS patients have also previously demonstrated elevated expressions of P-selection (an activated platelet cell surface receptor), increased lymphocyte-platelet complexes, and higher maximal platelet aggregation [12,13,14]. Similarly, patients with PAD have demonstrated significantly higher adenosine diphosphate (ADP) stimulated P-Selectin expression, bound fibrinogen, and maximal platelet aggregation when compared to healthy controls [15,16,17,18,19]. Past studies have also noted that platelet activation increases with increasing severity of PAD [20].

Platelets can affect the whole life cycle of atherosclerotic disease [21], from the early events of fatty streak formation to the final events of atherosclerotic plaque rupture and thrombus formation. As such, the medical management of atherosclerotic diseases often requires the inhibition of platelet activity, for which patients are often prescribed antiplatelet therapies such as acetyl salicylic acid (more commonly known as aspirin), which plays a vital role in preventing adverse cardiovascular events and decreasing thrombotic risk [22,23,24,25,26,27].

## 2. Aspirin as an Antiplatelet Therapy

Due to the high platelet reactivity and increased risk of adverse cardiovascular and thrombotic events, the prescription of an antiplatelet therapy is vital for patients with PAD and CAS [25,28]. The Antiplatelet Trialist Collaboration conducted a study found that a daily low dose of aspirin (75–150 mg) prevented adverse cardiovascular events by almost 25% in patients [29].

Aspirin obtains its antiplatelet effects through the inhibition of arachidonic acid mediated platelet aggregation. In short, aspirin irreversibly binds to the Cytochrome C Oxidase I (COX-1) enzyme within platelets, preventing the conversion of arachidonic acid, a fatty acid cleaved from the platelet membrane, into Thromboxane A2 (TxA2), a highly prothrombotic lipid mediator that leads to the activation and aggregation of platelets (Figure 2) [8,30,31].

Medical societies such as the American Heart Association and Society of Vascular Surgery, recommend a daily dose of aspirin, between 81 to 325 mg, to PAD and CAS as means of secondary prevention against adverse events such as MI, CVA, and cardiovascular-related death [25,26,28]. Although other antiplatelets such as clopidogrel have shown a similar, if not better protection against adverse events (as demonstrated in the CAPRIE trial), aspirin still remains the first line therapy for this patient population [32,33]. This may be as aspirin is far cheaper, more widely available, and has a lower cost per quality-adjusted life year gained (QALY), making it a more appealing therapy when compared to its drug counterparts [34].

## 3. Aspirin Resistance

Despite aspirin being the gold-standard antiplatelet therapy, a new phenomenon known as aspirin resistance has emerged within the scientific literature. Although not clearly outlined, it entails two broad definitions: (1) a “clinically defined aspirin resistance”—in which the standard dose of aspirin does not prevent recurrent adverse cardiovascular events, and (2) a “laboratory defined aspirin resistance”—in which the standard dose of aspirin does not produce the intended or normal effect that would be expected on platelet function tests when a patient is taking an antiplatelet, such as increased bleeding time, reduced TxA2 synthesis, or inhibited platelet aggregation [35,36,37,38,39,40].

Aspirin resistance can also be characterized into two categories: pharmacokinetic resistance, and pharmacodynamic resistance. Pharmacokinetic resistance occurs when a sufficient concentration of aspirin is unobtainable in the blood, and is therefore unable to completely inhibit thromboxane production and platelet aggregation. This can occur due to reduced aspirin absorption within the gut, increased aspirin degradation, and high platelet turnover, among other reasons. Pharmacodynamic resistance on the other hand occurs when there is a sufficient concentration of aspirin in the blood, however due to genetic polymorphisms within the COX-1 enzyme, or changes within the structure of platelets themselves, aspirin is unable to inhibit COX-1 and consequently, platelet aggregation. Notably, recent studies have found a COX-1 polymorphism that may increase the risk of aspirin resistance [41], as well as an anion efflux plump Multidrug Resistance Protein 4 (MRP4), which may contribute to reduced inhibition of COX-1 activity in patients [42].

It has been documented that ~20–30% of patients are resistant to their aspirin therapy, with some studies determining resistance to be as high as 60% in their respective patient population [36,43,44,45,46,47,48,49,50,51]. Recent studies suggest that aspirin resistance increases the risk of adverse cardiovascular events by almost three-fold in various patient populations [38,46,52]. This is supported by additional studies demonstrating a significant increase in the risk of developing MI, CVA and death in aspirin resistant patients. For instance, a study conducted by Chen et al. reported that aspirin resistant cardiovascular disease patients had a threefold higher rate of adverse events when compared to patients who aspirin sensitive [53]. Similarly, Krasopoulos et al. reported an odds ratio of nearly 6.0 for increased mortality in resistant vascular disease patients [54].

## 4. Antiplatelet Resistance Function Tests

Several antiplatelet function tests are currently being used both in clinical and laboratory settings for the testing of aspirin resistance.

### 4.1. Light Transmission Aggregometry (LTA)

Light transmission aggregometry (LTA) is the current gold-standard platelet function test and is the most widely used for aspirin resistance testing [55]. Whole blood is collected from patients of interest and separated in two types of plasma samples: platelet-rich plasma (PRP) and platelet-poor plasma (PPP), named according to the concentration of platelets within the plasma. Light is transmitted through both plasma samples and detected by a photocell on the opposite side [56]. PRP is given a baseline light transmission of 0%, and PPP is assigned as 100% light transmission [57]. A platelet agonist of interest is added to the PRP sample, and platelet aggregation is initiated. As platelets begin to activate and stick together into larger platelet aggregates, the PRP becomes more translucent, allowing for an increase in light transmission through the plasma (Figure 3). An increase in aggregation is recorded as a decrease in absorbance [58]. In LTA analysis, light transmission is proportional to platelet aggregation and hence can provide the total percentage of platelets aggregated in response to the platelet agonist of interest. LTA can be a useful method for invitro platelet testing, as a wide range of platelet activation pathways can be investigated using different agonists such as arachidonic acid, ADP, and thrombin, and plasma samples can be incubated with drugs of interest before testing in order to study the effects of these drugs on platelet aggregation [51,58]. Therefore, in patients taking antiplatelets, aggregation is expected to remain around 0%, and those who are not taking any antiplatelet therapy, the light transmission is expected to increase to approximately 100% [59,60]

Numerous studies have established ≥20% aggregation in response to arachidonic acid by LTA as a diagnostic cut-off for aspirin resistance [61,62,63,64,65,66,67]. Previous studies have also established a link between patients with ≥20% aggregation and significant increase in the risk of cardiovascular and thrombotic events [38,54].

Although considered the gold standard, data on the accuracy and reproducibility of LTA is still contested. Several limitations are associated with LTA. For instance, results can be influenced by several parameters such as platelet count, hematocrit, and lipid concentrations in the blood [59,60]. Several studies have noticed varying accuracy and reproducibility of LTA, noting high short-term reproducibility, with long-term reproducibility being significantly lower [68,69]. Furthermore, the sensitivity of LTA testing has been shown to be dramatically low when diagnosing patients with platelet defects (73% false-negative rate) [70]. LTA is also labor intensive, and requires trained personnel to conduct. Thus, while LTA has been the leading methodology for aspirin sensitivity testing, it has been shown to produce inconsistent results. This further highlights the critical need for a new, clear, standardized, and reliable method for aspirin sensitivity testing.

### 4.2. Platelet Function Analyzer 100/200 (PFA 100/200)

Platelet Function Analyzer (PFA) is a point-of-care platelet function test that records the closure time of a small aperture by hemostatic platelet plugs in an environment that mimics an injury. It makes use of a cartridge in which a membrane with a small aperture is coated with platelet agonists such collagen and epinephrine. Whole blood is aspirated through this small aperture. As blood flows through the membrane, the platelet agonists activate platelets and a platelet plug forms, and closes the aperture. The time taken for the aperture to fully close (closure time) is recorded. Several cartridges are available, including collagen/epinephrine, and collagen/ADP coated cartridges. Patients on antiplatelet therapies have increased closure times on cartridges related to the antiplatelets mechanism of action. For instance, patients on aspirin have a higher closure time on the collagen/epinephrine cartridge, whereas patients on clopidogrel will have a higher closure time on the collagen/ADP cartridge. An agreed upon closure time of >161 s on the epinephrine/collagen cartridge serves as the established cut-off value for diagnosing aspirin resistance [71,72].

While the PFA is convenient and more rapid in generating results when compared to other methods, concerns exist about its ability to determine aspirin resistance in patients. Crescente et al. conducted a systemic review suggesting that despite the increase in the use of PFA for aspirin resistance testing, confounding variables are generally not accounted for which can significantly affect the sensitivity for aspirin resistance diagnosis [73]. Furthermore, a study done by Lordkipanidzé et al. assessing aspirin resistance in coronary artery disease (CAD) patients reported that PFA over-estimated the percentage of patients who are aspirin resistant when compared to other platelet function tests such as LTA. It was also recorded that PFA, when compared to LTA, had poor correlation regardless of the agonist used [66]. Due to the significant confounding factors that may reduce closure time, PFA is prone to generating false positives within the patient population being investigated. Therefore, a different technology may provide a better, more efficient, and reliable tool for aspirin resistance.

### 4.3. Thromboelastography (TEG)

Thromboelastography (TEG) and Rotational TEG (ROTEM) are ex-vivo hemostatic assays that can provide real-time assessment of viscoelastic clot strength in whole blood [74]. Both assays utilize a needle suspended by a torsion wire in whole blood incubated in a heated cup, which continually oscillates. A blood coagulation agonist, such as tissue factor, or a platelet agonist such as ADP is added to the blood to initiate the formation of a hemostatic plug on the wire. As the plug forms, and the strength of the plug increases, more rotational torque is detected by the needle and transmitted to the wire. This torsion is interpreted by the TEG software to generate sophisticated plug formation/dissolution kinetics and determine plug strength [74,75].

Most studies conducted using TEG have focused on platelet reactivity rather than antiplatelet resistance. Only a few studies have recorded the effectiveness of TEG as a tool for predicting aspirin resistance in CAD, but no recent publications have examined its ability to predict aspirin resistance in patients with PAD [76,77]. Tantry et al. reported in 2005 that aspirin resistance tended to be rare in compliant patients, with only around 0.4% showing higher coagulation times after aspirin administration using TEG [78]. Further research on TEG/ROTEM is required to determine their capability of detecting aspirin resistance. Many different coagulation agonists can be used in TEG/ROTEM system, and investigating arachidonic acid may be a better option to detect aspirin resistance. A cut-off value for aspirin resistance is still to be determined for this assay.

### 4.4. Thromboxane B_2_ (TXB_2_) Testing

As established, aspirin inhibits the production of TXA_2_ by irreversibly acetylating the COX-1 enzyme. After its production, TXA_2_ is rapidly degraded into a more stable form, Thromboxane B_2_ (TXB_2_) [79]. Hypothetically, as aspirin reduces the production of TXA_2_, patients who are resistant to aspirin would have higher TXB_2_ production as compared to those who are sensitive. Previous studies have suggested that measuring serums TXB_2_ levels may be an effective method of determining aspirin resistance. As of date, no studies have demonstrated a strong correlation between aspirin resistance and increased levels of TXB_2_ in serum, and show poor agreement with other established aspirin resistance tests [79,80,81]

Similar to serum TXB_2_, urinary 11-dehydro thromboxane B_2_ (11-DTB_2_) is a stable metabolite of TXA_2_ which can be detected in urine. In 2002, Bruno et al. demonstrated that aspirin use was significantly associated with reduced 11-DTB_2_ levels in urine [82]. However in a later study conducted by the same group, the researchers demonstrated there was no difference between in 11-DTB_2_ levels when comparing differing aspirin doses, suggesting it may not be suitable for aspirin resistance testing [83]. Contrarily, urine analysis by Eikleboom et al. of 5529 patients taking aspirin demonstrated that higher levels of 11-DTB_2_ significantly predicted the future risk of myocardial infarction or cardiovascular death [84]. Further studies to establish a cut-off value for this metabolite as a method for diagnosing patients as aspirin resistant is required.

## 5. Potential In Vitro Tests for Aspirin Resistance

The aforementioned methods for aspirin resistance testing are associated with several challenges as they require a laboratory setting, well-trained personnel, and expensive equipment. A reliable, inexpensive, quick point-of-care test for diagnosing aspirin resistance that can be conducted at the bedside or in a small family care setting would facilitate more widespread testing. Two potential points-of-care tests for aspirin resistance have recently been introduced, Verify Now and Platelet Works.

### 5.1. Verify Now

Verify Now is a test for platelet aggregation using optical density to determine the amount of platelet aggregation in response to a platelet agonist. Whole blood from patients can be incubated with a platelet activation agonist of interest attached to microbeads. As the platelets aggregate, they attach to available fibrinogen-coated microbeads and aggregate out of solution, reducing the optical density of the blood. The change in absorbance is used to calculated a reading in Aspirin Reaction Units (ARU) [85].

A cut-off of >550 ARU is often the suggested value by the manufacturing company for diagnose aspirin resistance using arachidonic acid cartridge [86,87,88,89]. Verify Now has been approved by the Food and Drug Administration (FDA) as a suitable test for antiplatelet resistance for not only aspirin, but other antiplatelets as well, such as clopidogrel, prasugrel and ticagrelor [90]. Verify Now has proven to be a front-runner for point-of care aspirin resistance testing, showing a strong reproducibility and high specificity for aspirin resistance when LTA is used as a gold standard [88], and has also shown the highest predictive accuracy for predicting atherothrombotic events compared to other platelet function tests [91]. Additional research however is still required to deem its effectiveness in different vascular disease patient populations. Specifically, large cohort studies are required to validate and standardize cut-off points and correlations between antiplatelet resistance and adverse cardiovascular events.

### 5.2. Platelet Works

Platelet Works is a novel platelet function test that allows common hematology analyzers to detect platelet aggregation in whole blood in response to different agonists. It allows for a quick and accessible test by determining the platelet count before and after the addition of a platelet agonist. With these values, the percentage of platelet aggregation can be calculated [92]. Previous research suggests that Platelet Works correlates well with gold standard LTA analysis and may predict adverse cardiovascular events in antiplatelet resistant patients [85,93]. However, a recent study published by Anaya et al. comparing PFA-100 with Platelet Works recorded weak correlation between the two. Notably, the author did not investigate LTA in their study [94]. Thus far, only one study has sought to determine a cut-off value for diagnosing patients with aspirin resistance using Platelet Works [93]. They determined a cut-off of >52% maximal platelet aggregation using Platelet Works could predict aspirin non-sensitivity with a sensitivity and specificity of 91% and 69%, respectively. Further studies are required to validate this cut-off point, and its ability to predicts adverse cardiovascular events in aspirin resistant patients.

Thus, a well-supported, established, and consistent method for the diagnosis of aspirin resistance is still needed. Since personalized medical approaches are the new forefront in medical management, a similar approach is likely needed to overcome antiplatelet resistance, as each patient has a unique pharmacokinetic/pharmacodynamic profile and response to antiplatelet therapies.

## 6. Alternative Antiplatelet Therapies for Aspirin Resistant Vascular Patients

Several antiplatelet therapies are available for patients with vascular disease, however aspirin remains to be the most commonly prescribed in vascular medicine [25,28]. Furthermore, a strong alternative candidate to aspirin in patients with aspirin resistance is yet to be determined. Several studies within patients with coronary artery disease have been conducted, but they have not been reproduced within PAD/CAS patients. Table 1 outlines studies (since 2010) that investigated alternative treatment for aspirin resistance vascular patients.

### 6.1. Clopidogrel

Clopidogrel is a commonly prescribed antiplatelet therapy to patients with PAD and CAS. By irreversibly binding to the P2Y(12) receptor, it inhibits ADP-induced platelet activation and aggregation [104]. As previously mentioned, the Clopidogrel vs. Aspirin in Patients at Risk of Ischemic Events (CAPRIE) trial demonstrated that the long-term administration of clopidogrel was more effective at preventing adverse cardiovascular events such as MI, CVA and cardiovascular related death when compared to aspirin [33]. Though clopidogrel has demonstrated strong efficacy in preventing adverse cardiovascular events, resistance to clopidogrel has been demonstrated to be almost as common as aspirin resistance. Similar to aspirin resistance, no clear definition has been clinically accepted, but several studies have demonstrated resistance in 5–63% of patients [105,106,107,108,109,110,111].

### 6.2. Ticareglor

Ticareglor, another P2Y(12) receptor inhibitor has also been suggested as an alternative treatment for patients with aspirin resistance. The Effect of Ticagrelor on Health Outcomes in Diabetes Mellitus Patients Intervention (THEMIS) trial demonstrated that ticagrelor, in combination with aspirin was significantly more effective at preventing adverse cardiovascular events when compared to aspirin alone [112]. Ticagrelor has also shown a significantly lower percentage of resistance (between 0–3%) when compared to its counterpart clopidogrel [113]. One study demonstrated that 100% of patients that were resistant to aspirin were sensitive to ticagrelor, demonstrating its potential as an alternative antiplatelet therapy [51]. However, ticagrelor has not been approved for patients with vascular disease. Therefore, further studies on its efficacy in patients with vascular diseases is needed.

### 6.3. Rivaroxaban

A recent breakthrough study, the Cardiovascular Outcomes for People Using Anticoagulation Strategies (COMPASS) trial, demonstrated that the addition of 2.5mg rivaroxaban to daily aspirin therapy significantly reduced the risk of recurring adverse cardiovascular events in patients with stable cardiovascular disease [97]. There was, however, an associated increased risk in major bleeding in those taking dual therapy. In vitro studies have also demonstrated that rivaroxaban and aspirin in combination significantly reduced arachidonic acid induced platelet aggregation by LTA, when compared to either alone [101]. With the addition of low dose (2.5 mg) rivaroxaban ex vivo, aspirin resistance was overcome in 58% of the patients tested [101]. However, further in vivo studies are needed to confirm these findings.

Several Chinese medical herbs have been suggested as alternative treatments for patients with aspirin resistance [114,115]. For example, a study conducted by Liao et al. in a zebrafish vascular disease model demonstrated that leonurine, a chemical isolated from Leonurus japonicus, a flowering plant, reduces platelet activity. Several other studies also have demonstrated Chinese herbal medicines for potential use in the treatment of aspirin resistance [116]; however with a small sample size and limited data on humans,, further research is required.

## 7. Conclusions

Aspirin is currently the gold standard antiplatelet therapy for the prevention of secondary adverse cardiovascular events in patients with atherosclerotic disease and remains the most often prescribed antiplatelet. While the prevalence of resistance to aspirin therapy is well established, point-of-care antiplatelet tests with strong correlation with other gold standard tests and high predictability of adverse events are still lacking. As of date, inconsistencies are present with regards to the methodologies used to diagnose aspirin resistance, and little to no standardization of diagnostic thresholds. Verify Now and Platelet Works are promising new assays that could detect aspirin resistance in an inexpensive, point-of-care manner. These technologies may be utilized by physicians at the bedside of to provide personalized antiplatelet therapy. The convenience and functionality of these tests is evident; however, additional research is required to strongly establish a cut-off value for aspirin resistance that can predict future adverse cardiovascular events. Similarly, further research is required to determine the efficacy of alternative treatments, such as clopidogrel, ticagrelor, rivaroxaban, cilostazol, traditional herbs, etc., in overcoming aspirin resistance among vascular patients.

## Figures and Tables

**Figure 1 ijms-23-11317-f001:**
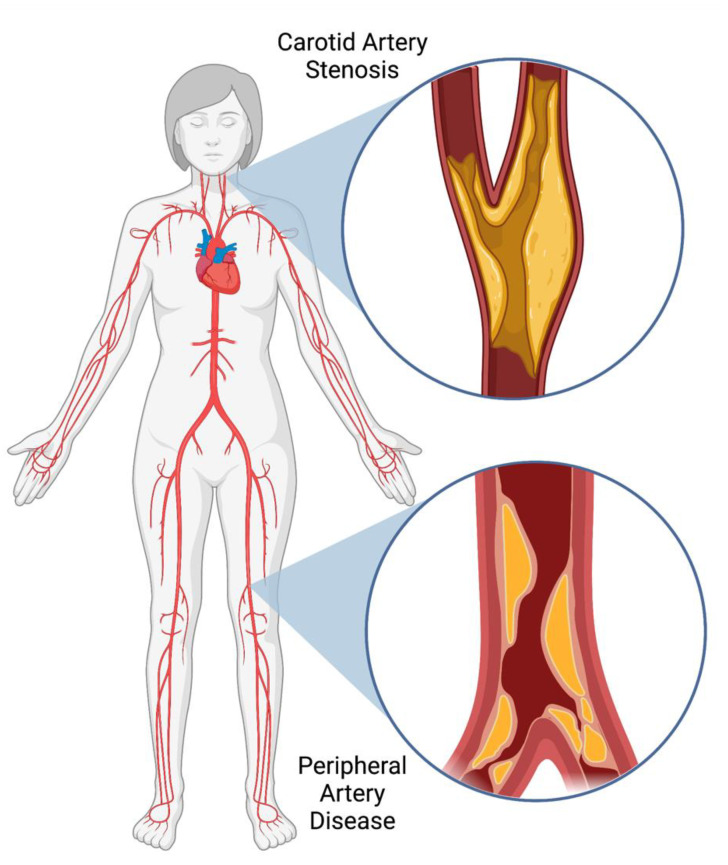
Example of the anatomical locations of atherosclerotic plaque formation other than the coronary vessels such as in carotid artery stenosis (upper diagram), and peripheral arterial disease (lower diagram).

**Figure 2 ijms-23-11317-f002:**
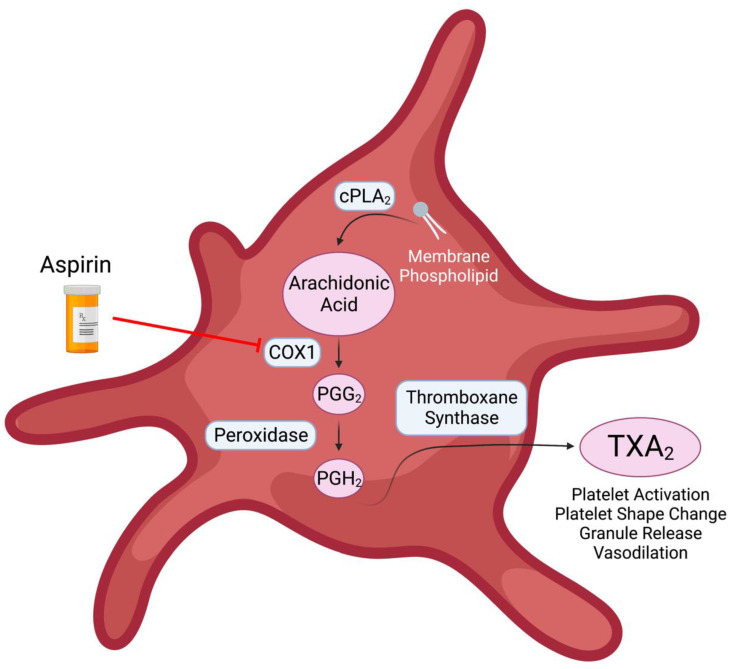
Overview of the inhibition of the Cytochrome c oxidase I (COX-1) by aspirin in platelets.

**Figure 3 ijms-23-11317-f003:**
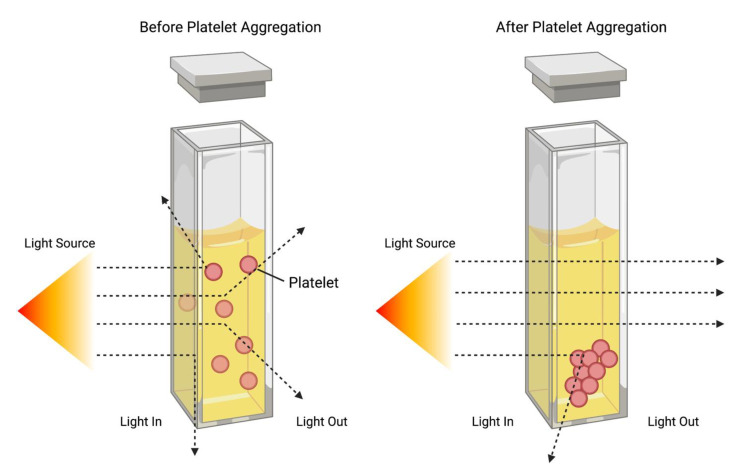
Comparing light transmission before and after platelet aggregation using light transmission aggregometry. Light rays from the light source are represented with the dotted line.

**Table 1 ijms-23-11317-t001:** Studies investigating alternative therapies in patients with aspirin resistance. MI: myocardial infarction; CI: confidence interval; TIA: transient ischemic attack; MACE: major adverse cardiovascular event.

Author	Title	Study Design	Purpose	Target Population	Outcome
Aggarwal et al., 2022 [95]	P2Y_12_ inhibitor versus aspirin monotherapy for secondary prevention of cardiovascular events: meta-analysis of randomized trials	Meta-analysis of randomized control trials	P2Y12 inhibitor versus aspirin in reducing adverse cardiovascular events	Peripheral arterial disease	When compared to aspirin, P2Y12 inhibitors reduced MACE by 11% (0.89, 95% CI 0.84–0.95, I2 = 0%) and MI by 19% (0.81, 95% CI 0.71–0.92, I2 = 0%).
Bonaca et al., 2020 [96]	Rivaroxaban in peripheral artery disease after revascularization	Double blind, control trial	Low dose 2.5 mg rivaroxaban twice daily plus daily low dose aspirin versus aspirin alone in reducing adverse cardiovascular events	Peripheral arterial disease	Low dose 2.5 mg rivaroxaban twice daily plus aspirin significantly reduced the incidence of acute limb ischemia, major amputation, MI, stroke or cardiovascular death when compared to aspirin alone (hazard ratio, 0.85, 95% confidence interval [CI], 0.76 to 0.96; *p* = 0.009)
Eikelboom et al., 2017 [97]	Rivaroxaban with or without aspirin in stable cardiovascular disease	double blind trial	Low dose 2.5 mg rivaroxaban twice daily plus daily low dose aspirin versus aspirin alone for secondary cardiovascular prevention	Atherosclerotic vascular disease	Cardiovascular death, stroke, or myocardial infarction was reduced in patients taking both low dose rivaroxaban and low dose aspirin compared to aspirin alone (hazard ratio, 0.76; 95% confidence interval [CI], 0.66 to 0.86; *p* < 0.001; z = −4.126)
Gengo et al., 2016 [98]	Platelet response to increased aspirin dose in patients with persistent platelet aggregation while treated with aspirin 81 mg	Retrospective, noninterventional study	Does increasing dosage for aspirin resistant patients reduce ischemic events	Neurovascular disease	After dose adjustment, there was a significant reduction in recurrent stroke, TIA, or TIA-like symptoms (*p* < 0.001).
Johnston et al., 2016 [99]	Ticagrelor versus aspirin in acute stroke or transient ischemic attack	Double blind, control trial	Ticagrelor versus aspirin in reducing adverse cardiovascular events	Carotid artery stenosis	There was no significant difference between aspirin and ticagrelor in reducing the rate of stroke, myocardial infarction, or death at 90 days.
Khan et al., 2020 [100]	Personalization of aspirin therapy ex vivo in patients with atherosclerosis using light transmission aggregometry	Ex vivo single center pilot study	Does aspirin dose adjustment reduce aspirin resistance ex-vivo	Peripheral arterial disease	Of 9 patients resistant to aspirin, 100% of patients became sensitive at higher doses
Khan et al., 2022 [101]	Low-dose aspirin and rivaroxaban combination therapy to overcome aspirin non-sensitivity in patients with vascular disease	Ex vivo single center pilot study	Does the addition of low dose 2.5 mg rivaroxaban to low dose aspirin reduce aspirin resistance ex-vivo	Peripheral arterial disease	Rivaroxaban reversed aspirin resistance in 11 of 19 resistant patients (58%)
Lee et al., 2010 [102]	Addition of cilostazol reduces biological aspirin resistance in aspirin users with ischemic stroke: a double-blind randomized clinical trial	Multi-center, double blind, control trial	Does the addition of 100 mg cilostazol twice daily to daily low dose aspirin reduce the incidence of aspirin resistance	Carotid artery stenosis	Incidence of aspirin resistance was not significantly different between aspirin plus cilostazol versus aspirin alone
Xingyang et al., 2017 [103]	Platelet function-guided modification in antiplatelet therapy after acute ischemic stroke is associated with clinical outcomes in patients with aspirin nonresponse	Multi-center, retrospective study	Does modifying antiplatelet therapy (increase dosage, or alternative antiplatelet such as clopidogrel or cilostazol) in aspirin resistant patients reduce ischemic events	Carotid artery stenosis	Antiplatelet therapy modification reduced ischemic events (hazard ratio, 0.67; 95% confidence interval [CI], 0.62–0.97; *p* = 0.01)

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
