# Peer review of "Aspirin Resistance in Vascular Disease: A Review Highlighting the Critical Need for Improved Point-of-Care Testing and Personalized Therapy"

_ijms, 2022, doi:10.3390/ijms231911317_

Round 1
Reviewer 1 Report
This review summarized the recent advances about antiplatelet resistance in vascular disease. This review paper was not well written. The content should be well organized and presented. The lack of deep analysis and summary is a major weakness.
The followings are a few examples:
1. The "Introduction" section is too long. This section could be more concisely written to bring focus to elements discussed in the review.
2. "Platelets' Role in Atherosclerosis" could be shortened and merged into "Introduction" section.
3. The overall quality of the figures must be improved. The font size in Figure 2 is too small to read.
Author Response
Thank you very much for your constructive feedback. We have addressed your comments and made some changes to the manuscript. We hope these changes will increase the quality of our review. Please see below for our responses to your comments.
1. & 2. The "Introduction" section is too long. This section could be more concisely written to bring focus to elements discussed in the review. "Platelets' Role in Atherosclerosis" could be shortened and merged into "Introduction" section.
This section has been merged into the introduction, and also significantly reduced to introduce the topic covered. Please see Pages 2-3, lines 30-95
3. The overall quality of the figures must be improved. The font size in Figure 2 is too small to read.
The figures have been edited to contain more information, and the font sizes have been increased for ease of reading, please see Pages 2-3 Figures 1 and 2.
Reviewer 2 Report
The manuscript submitted from Hamzah Khan et al., entitled "Antiplatelet Resistance in Vascular Disease: A Review Highlighting the Critical Need for Improved Point-of-Care Testing and Personalized Therapy" is interesting, however the authors should improve some critical points: - The authors should insert a table that lists all previous studies focused on drug therapy alternative to aspirin. - The authors did not mention preclinical model used to identify the specific therapy. For example there are different mice and more recently zebrafish models that have been used to test new molecules in the context of aspirin resistance in vascular diseases. - The authors should also include a future perspective in the way of drug discovery. This points should be improved to increase the novelty and quality of the manuscript.Author Response
Thank you very much for your review and your feedback on our manuscript. You have provided excellent feedback which we highly appreciate. Please find our comments and adjustments to our manuscript to your suggestions below.
1. The authors should insert a table that lists all previous studies focused on drug therapy alternatives to aspirin
Please find the suggested table in the manuscript (Table 1), Pages 8-10 Line 685.
2. The authors did not mention preclinical model used to identify the specific therapy. For example there are different mice and more recently zebrafish models that have been used to test new molecules in the context of aspirin resistance in vascular diseases.
Thank you for this great suggestion, please find the addition of this suggestion on Page 11 lines 755-761.
3. The authors should also include a future perspective in the way of drug discovery.
We have included some text in the manuscript addressing the future perspectives in drug discovery, please see Page 19, lines 937-940.
Round 2
Reviewer 1 Report
The authors have adequately addressed the reviewer's comments. I have no further comments to suggest.
Reviewer 2 Report
The authors satisfied all my concerns.